# Noise amplification and ill-convergence of Richardson-Lucy deconvolution

Yiming Liu [1,2], Spozmai Panezai[1,2], Yutong Wang[1,2] & Sjoerd Stallinga [1] ✉

Richardson-Lucy (RL) deconvolution optimizes the likelihood of the object estimate for an incoherent imaging system. It can offer an increase in contrast, but converges poorly, and shows enhancement of noise as the iteration progresses. We have discovered the underlying reason for this problematic convergence behaviour using a Cramér Rao Lower Bound (CRLB) analysis. An analytical expression for the CRLB diverges for spatial frequency components that approach the diffraction limit from below. The resulting mean noise variance per pixel diverges for large images. These results imply that a regular optimum of the likelihood does not exist, and that RL deconvolution is necessarily ill-convergent.

An important quest in the field of imaging is to devise instruments and methods to deliver the sharpest and most contrast rich images possible. Computational enhancement of raw images acquisitions are an important and broadly applied inroad to increase sharpness and contrast. This enhancement can be achieved via image processing steps such as filtering operations that are applied ad hoc, agnostic to the underlying physics of the image formation[1]. Current data driven approaches rely largely on learning algorithms[2,3]. Deconvolution refers to a family of algorithms that are based on estimating the underlying object, using statistical inference and a model of the image formation[4,5]. The archetypical algorithm in this field is Richardson–Lucy (RL) deconvolution[6,7]. An important advantage of RL deconvolution is that it enables reconstruction of out-of-band information, depending on the type of object that is imaged[8].

The application of RL deconvolution to practical imaging settings in astronomy or microscopy has brought to light that the algorithm converges slowly, if at all. Moreover, with increasing number of iteration steps an apparent noise structure builds up, originating from small perturbations of the input due to physical noise and/or numerical noise[4,8]. Essentially two different mitigation strategies have been followed to deal with this problematic behavior. The first strategy is to stop the iteration prematurely, in an ad-hoc manner, or derived from additional metrics[9], instead of a tolerance criterion derived from a numerical analysis of the convergence of the procedure. The second strategy is based on the use of prior information. In case the to-be estimated object is known or expected to have certain features, then these features can be incorporated into the merit function that is

numerically optimized. Such priors could for example be based on standard L2 or L1 norm metrics[10,11].

The underlying question why RL deconvolution has a problematic convergence or why the procedure is so sensitive to noise has not been answered in the half century since its inception. Zaccheo et al. studied the Cramér Rao Lower Bound (CRLB) for RL-deconvolution numerically, via computation of linear restoration weights, but did not derive the Fisher matrix or provide an analytical treatment, and made no connection to noise amplification and convergence behavior[12]. Here, we address the questions of ill-convergence and noise amplification of RL deconvolution by deriving and verifying experimentally an analytical expression for the CRLB. As the RL algorithm is a form of Maximum Likelihood Estimation (MLE) for the ground truth object, a hypothetical well behaved optimum must have a lower bound on the precision of the estimate of the object, and this lower bound is the CRLB. We will prove that the CRLB diverges and that hence the original assumption of a regular, well-behaved optimum must be false. Our result also intimately connects noise sensitivity and noise amplification to a lack of convergence of the iterative procedure.

## Results

### Cramér Rao lower bound

Consider an incoherent optical imaging system, and suppose we take an image $n = (n_1, \ldots, n_K)$ with pixels $j = 1, 2, \cdots, K$, where the pixel values have been corrected for camera gain and offset to represent the measured photon count. The expected photon count is $\mu = (\mu_1, \ldots, \mu_K) = gx$, where $x = (x_1, \ldots, x_K)$ represent the underlying

[1]Department of Imaging Physics, Delft University of Technology, Delft, The Netherlands. [2]These authors contributed equally: Yiming Liu, Spozmai Panezai, Yutong Wang. ✉e-mail: s.stallinga@tudelft.nl

object, and where the (shift invariant) matrix $g$ represents the Point Spread Function (PSF). The Fourier Transform of $g$ is the Optical Transfer Function (OTF) $\hat{g}$, and is zero for spatial frequencies $|\vec{q}| \geq 2NA/\lambda$, where $NA$ is the Numerical Aperture of the imaging lens and $\lambda$ the imaging wavelength. The image $n$ and the expected image $\mu$ differ by shot noise and readout noise with standard deviation $\sigma$ (mixed Poisson-Gauss statistics). RL deconvolution provides an estimate for the ground truth object $x$.

We have found an analytical expression for the Cramér Rao Lower Bound (CRLB) of this estimate (see Supplementary Information for a full derivation):

$$\left\langle \left| \Delta \hat{x}_k \right|^2 \right\rangle \geq K \frac{\mu_{av} + \sigma^2}{\left| \hat{g}_k \right|^2} \tag{1}$$

Here $\hat{x}_k$ is a Fourier-component of the object estimate within the OTF support, that is for spatial frequency $|\vec{q}_k| < 2NA/\lambda$, and $\Delta\hat{x}_k$ represents the standard deviation over all noise realizations of the imaging experiment. The parameter $\mu_{av}$ is the expected number of photons per pixel, averaged across the image. The scaling of the lower bound with the inverse square of the OTF immediately implies that the noise variance in Fourier space must diverge for spatial frequencies $\vec{q}$ that approach the diffraction limit from below. The noise variance in real space, averaged over all pixels, satisfies the inequality:

$$\frac{1}{K} \sum_{j=1}^{K} \left\langle \left| \Delta x_j \right|^2 \right\rangle \geq (\mu_{av} + \sigma^2) \left( \frac{1}{K} \sum_{|\vec{q}_k| < 2NA/\lambda} \frac{1}{\left| \hat{g}_k \right|^2} \right) \tag{2}$$

The term between brackets in the lower bound is the average of the inverse square of the OTF, and is much larger than one. As the image size grows large ($K \to \infty$) this average becomes an integral, which formally diverges (see Supplementary Information).

These results have profound implications for RL-deconvolution. If the RL-deconvolution mathematically converges, then we know that it converges to a (local) optimum of the likelihood function. For this hypothetical optimum the CRLB theorem must hold. We have now found that the CRLB diverges for spatial frequencies approaching the diffraction limit. The conclusion of this reductio ad absurdum must therefore be that RL-deconvolution cannot converge. This matches with the behavior that is found in practice, where continued iteration always seems to lead to noise amplification. This makes sense, as we now have proven that the in-band noise must diverge when the likelihood optimum is approached.

The surprising conclusion is that the noise blowup is not rooted in the ill-posedness of the deconvolution problem for the spatial frequencies above the cutoff. Rather, the OTF gradually decreases with spatial frequency, and the noise amplification is strongest in the spatial frequency region approaching the cutoff. It is therefore also to be expected that noise amplification will occur in a hypothetical imaging system with a non-zero OTF for all spatial frequencies. The above derived CRLB would then be valid for all Fourier components of the object estimate, and noise amplification would occur for all spatial frequencies with small OTF. In particular, if the integral of the inverse squared OTF diverges, then the average noise variance across all pixels will still diverge in the limit of a large FOV. Another hypothetical case of interest is that of a flat OTF until the cutoff. Then, noise amplification and blowup is not expected as the CRLB is also flat in spatial frequency space. The applicability of RL deconvolution to this case, however, is questionable, as the flatness of the OTF does not induce the algorithm to retrieve object spatial frequencies with low contrast, i.e. no sharpening effect can be expected. Furthermore, the flat OTF violates the Lukosz bound and gives rise to a non-negative PSF[13,14], which is problematic in view of the built-in positivity constraint of the RL deconvolution algorithm.

## Experiment

We have tested the theoretical predictions by repeated noise independent imaging ($M = 10$ times) of mitochondria and actin structures using a widefield microscope (see Methods for details). A cropped region of the entire Field Of View of the $M$ noise independent raw images were fed into an RL deconvolution routine for 300 iterations. The intermediate estimates were stored, and the mean and variance of the Fourier transforms of the intermediate estimates were computed. Figure 1 shows the raw images and the deconvolution results for a number of different iteration steps, as well as reference high resolution images. Apparently, a limited number of iterations improves contrast and apparent resolution, a too large number of iterations gives rise to noise build-up. This is quantified by the mean and statistical spread over the $M$ statistically independent deconvolutions in real and in Fourier space shown in Figs. 2 and 3 and Supplementary Movie 1 and 2. There is a clear buildup of a noise structure, with a spatial frequency content that strongly peaks for the spatial frequencies just below the cutoff spatial frequency $2NA/\lambda$. This spatial frequency noise peak moves closer to the cutoff as the iteration progresses, in agreement with the expected asymptotic behavior according to the CRLB. The scaling of the noise variance for the limiting case of a very large number of iterations is with the predicted inverse square of the square of the OTF, as shown in Fig. 4a, b. There, we see the buildup of noise with iteration number, starting from the flat noise profile of the raw image data before the first iteration, which is well below the CRLB. The noise level must rise to at least the level of the CRLB when the likelihood increases to its maximum. The experimental curves indicate that this situation is gradually approached as the number of iterations grows, with a noise level at the CRLB for a gradually growing region in spatial frequency space.

## Impact of OTF

In the deconvolution of our experimental data we have used the OTF computed from a vectorial PSF model using aberration coefficients determined from a through-focus bead calibration experiment (see Methods for details). Supplementary Fig. 1 and Supplementary Movie 3 show the result of this analysis, indicating that the actual imaging conditions for the microscope configuration were aberrated with small but sizeable amounts of coma and spherical aberration. Inaccuracy of the estimated or assumed OTF of the microscope has impact on the outcome of the deconvolution, and the buildup of noise.

We have investigated this for our experimental data by computing additional deconvolutions, where we blurred the vectorial PSF with experimental aberration coefficients with a Gaussian function, for different values of standard deviation of the Gaussian. Also for this form of deliberately induced inaccuracy of the OTF, the noise variance scales with the inverse square of the OTF used in the deconvolution (Fig. 4c–f).

We found further evidence in simulation. We used a stock image, low-pass filtered it with a scalar diffraction OTF $\hat{g}_{sc}$, added Poisson distributed noise, and subsequently fed the result into the RL deconvolution routine with an OTF $\hat{g}_{dec} = \hat{g}_{sc}^{\,\nu}$, where $\nu$ is an exponent quantifying the inaccuracy of the OTF. For $\nu < 1$ the assumed OTF is better than the ground truth, leading to a sharper assumed PSF, for $\nu > 1$ the assumed OTF is worse than the ground truth, leading to a more blurred assumed PSF. In the latter case, the RL deconvolution is more aggressively trying to recover high spatial frequency content. Supplementary Movie 4 shows that this will result in a bit higher contrast, but with higher noise buildup and with the risk of edge ringing artefacts. It appears that the noise variance scales with the inverse square of the OTF used in the deconvolution, not with the actual OTF of the imaging system.

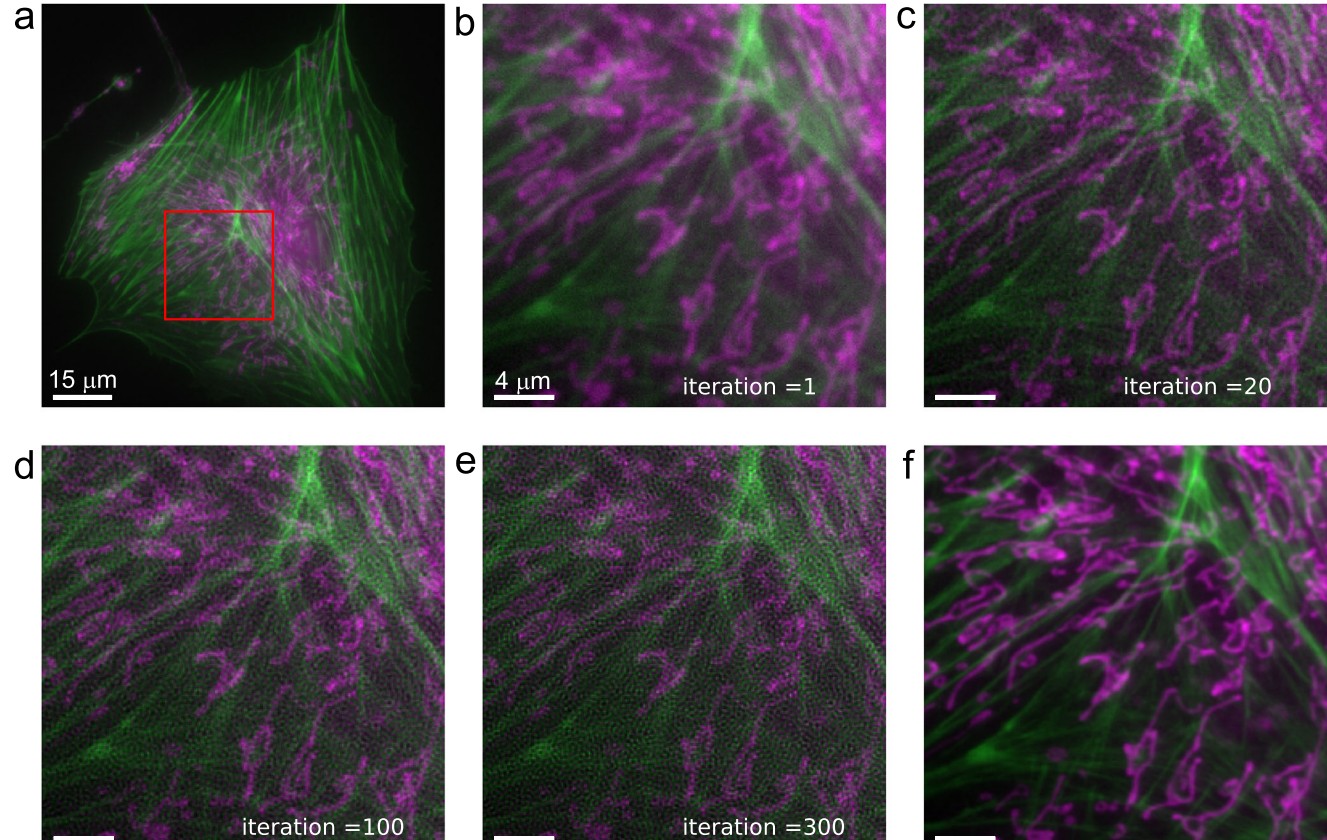

**Fig. 1 | Richardson-Lucy deconvolution of mitochondria (pink) and actin (green) in BPAE cells with 40×/NA0.95 objective lens. a** Raw image acquisition (large crop). **b**–**e** Deconvolution results for inset indicated in red in **a** for iteration numbers 1, 20, 100, 300 (iteration = 1 indicates raw image acquisition). **f** High resolution reference image taken with 100×/NA1.50 objective lens. Scale bar **a** 15 µm, scale bar **b**–**f** 4 µm. No repeated imaging/deconvolution ($M$ = 1).

## Gain in spectral signal to noise ratio

In the ultimate case where the number of iterations goes to infinity, there can be no gain in Spectral Signal to Noise Ratio (SSNR) compared to the original raw image, as rewriting the CRLB given in Eq. (1) gives:

$$SSNR_k \equiv \frac{|\hat{x}_k|^2}{\langle|\Delta\hat{x}_k|^2\rangle} \leq \frac{|\hat{g}_k|^2|\hat{x}_k|^2}{K(\mu_{av}+\sigma^2)} = \frac{|\hat{\mu}_k|^2}{K(\mu_{av}+\sigma^2)} \qquad (3)$$

where the bound on the right hand side represents the SSNR of the original raw image. For the intermediate cases of limited numbers of iterations, and for most practically occurring object structures, however, a real gain can be accomplished. Supplementary Figs. 2 and 3 show the increase in SSNR over the original widefield image acquisition for our experimental datasets, for varying exposure times, and for varying blurring factors of the OTF used in deconvolution. The spatial frequency where the SSNR gain peaks shifts to higher spatial frequencies when the iteration progresses, with a maximum gain obtained typically for several 10 s of iterations, in agreement with qualitative observation. It can also be observed that the SSNR gain increases with the SSNR of the starting point, the raw images. Apparently, the RL-deconvolution routine needs a basic SSNR level to pick up and amplify along the way, before its inevitable decline towards the SSNR of the starting point. Assuming a less sharp PSF in the RL-deconvolution, is another factor in improving the gain in SSNR that is obtained. It appears that SSNR can also be increased for spatial frequencies beyond the diffraction limit (see e.g. Supplementary Fig. 3j). This SSNR increase drops to zero when the number of iterations goes to infinity, similar to the SSNR increase for the spatial frequencies below the diffraction limit.

This points to a blowup of noise also for these higher spatial frequencies, as seems to be the case in experiment (see growth in noise level curves with iteration number in Fig. 4), even though a formal CRLB cannot be established in the spatial frequency region beyond the cutoff.

## Discussion

The key conclusion of the current paper is that noise amplification and ill-convergence are characteristics of RL deconvolution that cannot be avoided. That does not imply, however, that deconvolution methods have no practical use. Application of a restricted number of RL iterative steps does work on many if not most objects of interest that are imaged. A real gain in contrast and resolution can be achieved, proven by the increase in SSNR. Apparently, in the course of the iterative procedure, first signal at high spatial frequencies builds up and noise is amplified only later on. A detailed study of how noise propagates through the update steps of the algorithm might reveal how this difference between signal and noise arises.

The current work propels recent research lines focusing on noise properties of optical imaging, and how these could be exploited. Estimation of camera gain and offset from image noise alone was proposed in ref. 15. We have earlier investigated noise propagation in the image reconstruction of Structured Illumination Microscopy (SIM, a method for resolution improvement in incoherent optical imaging based on patterned illumination[16]), and demonstrated ways to optimize contrast or final noise profile[17]. Recently, an interesting inroad to determine after how many steps RL iteration is terminated has been proposed[18]. In this approach, use is made of the possibility to split the observed image in two noise independent halves[17,19]. Running the RL deconvolution algorithm on

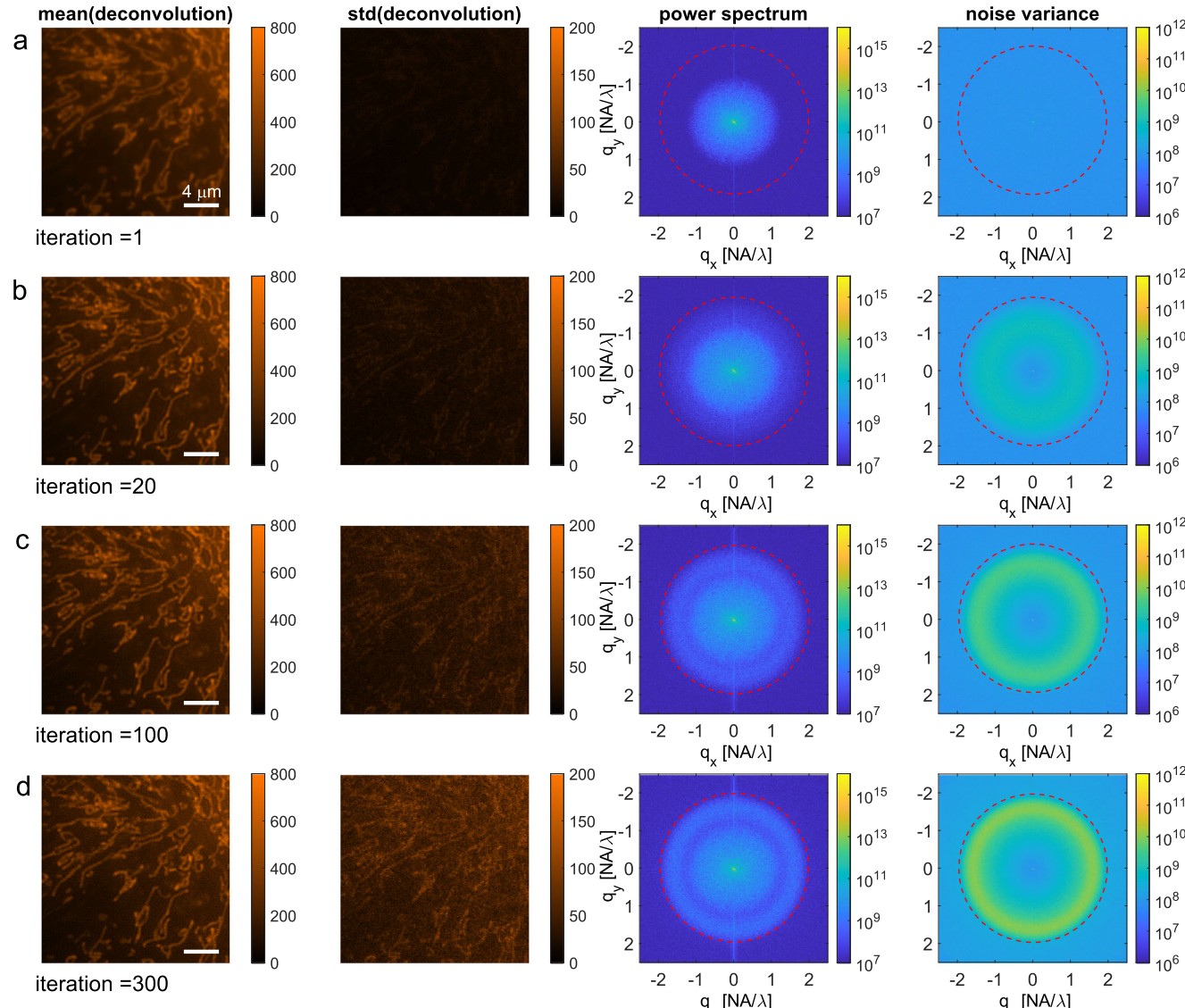

**Fig. 2 | Illustration of noise blow-up in RL-deconvolution of an image acquisition of mitochondria in BPAE cells. a–d** Rows indicate the iteration number, starting from the raw image acquisition (iteration = 1). Columns show the mean (first column) and standard deviation (second column) over the $M = 10$ noise independent deconvolutions, the mean square (third column) and the variance (fourth column) of the FT of the $M$ deconvolutions. The first two columns show a $251 \times 251$ pixel crop of the $960 \times 960$ pixel area on which the deconvolution is applied in order to make the smaller details more clear. The last two columns pertain to the full $960 \times 960$ pixel deconvolution. The diffraction limit at $2NA/\lambda$ is indicated with red dashed circles. Scale bar (**a–d**) 4 μm.

both halves and checking for the consistency of the two deconvolution results enables a more objective assessment of the optimum number of iterations.

Another inroad to build on our CRLB analysis is in the application of different Bayesian concepts to analyze convergence and noise sensitivity. In particular the Van Trees Inequality (VTI), which generalizes the CRLB to take into account prior information[20–22], could be used when priors are added to the to-be-optimized log-likelihood. A finite VTI is then a necessary condition for proper convergence of such deconvolution approaches.

Finally, we envision CRLB analyses of other object estimation methods in optical imaging. This applies to incoherent imaging techniques like SIM, where insight into the convergence of deconvolution approaches to SIM[23,24] could be gained, but also to coherent or partially coherent imaging techniques like Coherent Diffractive Imaging (CDI) and Fourier ptychography[25–27]. The numerical studies of refs. [28,29] into the CRLB for phase imaging could be placed on a theoretical basis with our approach.

## Methods
### Experimental setup
We have imaged the MitoTracker red channel (emission wavelength 599 nm) and Alexa Fluor 488 Phalloidin channel (emission wavelength 512 nm), labeling mitochondria and actin, respectively, of a sample of Bovine Pulmonary Arterial Endothelial cells (BPAE-cells, ThermoFisher FluoCells \#1), with a Nikon Ti-E microscope body with a 40×/NA0.95 objective lens with 1.5× magnification adapter, equipped with an sCMOS camera (Teledyne Kinetix, 3200 × 3,200 pixels, pixel size 6.45 μm, 1.93 e median readout noise determined from 5 sets of 1000 dark images). The back projected pixel size was 108 nm, which is oversampled compared to the Nyquist sampling distance of 158 nm for the red channel and 135 nm for the green channel. These parameters settings were selected in order to accommodate buildup of high spatial frequencies during the RL deconvolution without danger of running into possible aliasing artefacts. Exposure times of 380 ms (mitochondria/red) and 260 ms (actin/green) were chosen to guarantee a decent SSNR of the raw

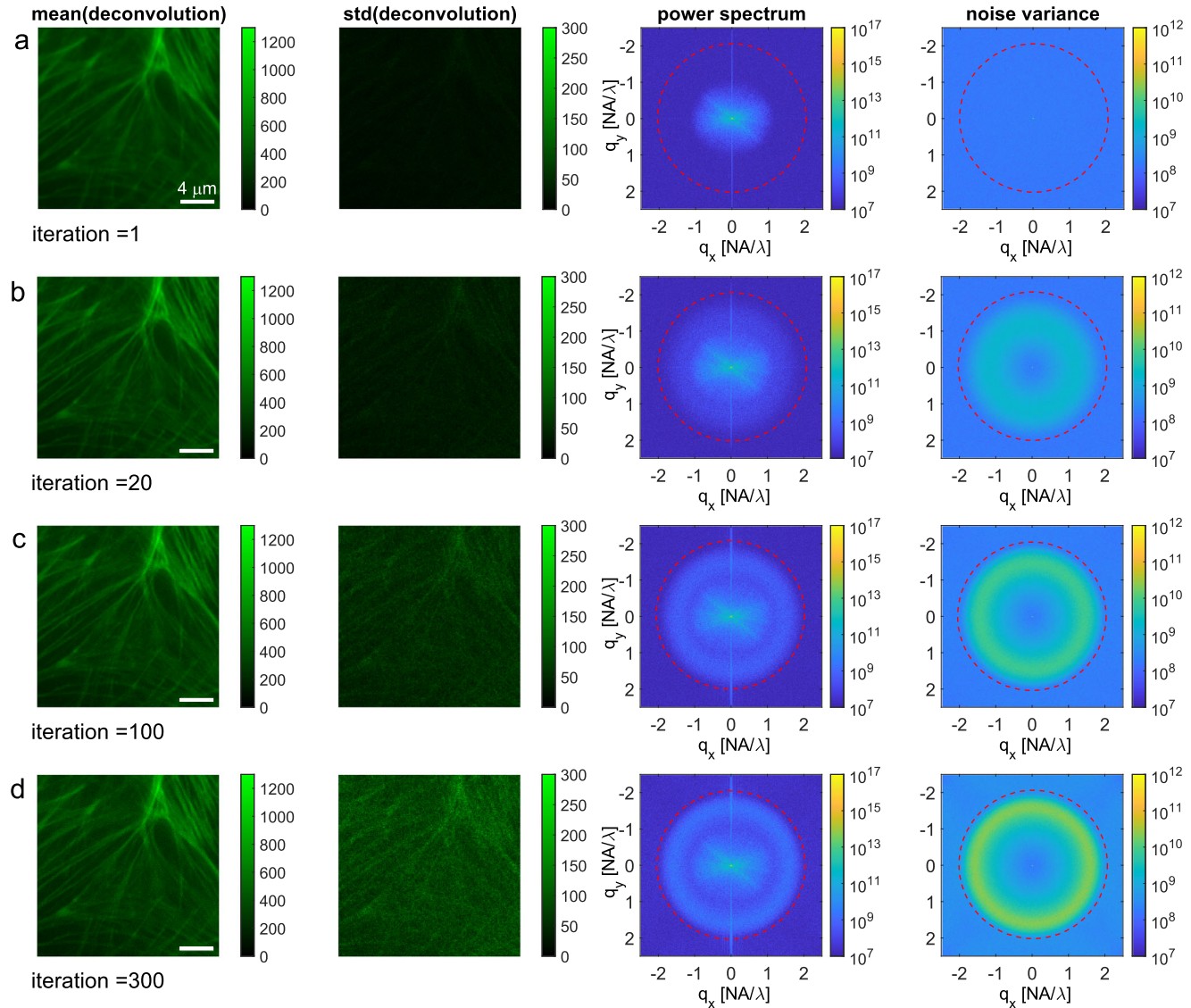

**Fig. 3 | Illustration of noise blow-up in RL-deconvolution of an image acquisition of actin in BPAE cells. a–d** Rows indicate the iteration number, starting from the raw image acquisition (iteration = 1). Columns show the mean (first column) and standard deviation (second column) over the $M = 10$ noise independent deconvolutions, the mean square (third column) and the variance (fourth column) of the FT of the $M$ deconvolutions. The first two columns show a $251 \times 251$ pixel crop of the $960 \times 960$ pixel area on which the deconvolution is applied in order to make the smaller details more clear. The last two columns pertain to the full $960 \times 960$ pixel deconvolution. The diffraction limit at $2NA/\lambda$ is indicated with red dashed circles. Scale bar 4 μm.

images. Additional exposure times of 20, 87, and 195 ms (mitochondria/red), and 10, 65, and 130 ms (actin/green) were used to study the impact of SSNR on the deconvolution outcome. Reference images for the RL deconvolution results were made with a $100\times$/ NA1.50 objective lens. The estimated gain and offset were $g = 3.0$ and $O = 100$, respectively, where the gain was estimated using the procedure of ref. 15 and the offset was determined from the set of dark images. For both color channels $M = 10$ noise independent acquisitions were made. A region of $960 \times 960$ pixels of the $M$ raw images were fed into the RL deconvolution routine for 300 iterations. The intermediate estimates were stored, and the mean and variance of the Fourier transforms of the intermediate estimates were computed. We used the default RL-algorithm, with a slight modification to take the Gaussian camera readout noise into account as well (see Supplementary Information). The impact of this was rather small, as the typical shot noise variance (mean number of photons per pixel) was usually much larger than the readout noise variance (around 4 based on the above mentioned 1.93 e we measured).

## OTF estimation

We have evaluated the optical quality by measuring a bead sample (TetraSpeck Microspheres, catalog number T7279, 100 nm diameter, green channel) through focus (100 nm focus step, 10 μm range). Regions of interest in the image data around beads were selected and for 65 beads a $21 \times 21$ area was cropped out. After gain and offset correction the image data was processed using the aberrations estimation procedure outlined in ref. 30. Zernike coefficients $A_{nm}$ up to second aberration order ($n + |m| \leq 2(j + 1)$ for aberration order $j = 2$), as well as the coefficient for third order spherical aberration $A_{80}$ were fitted. The mean of the aberration values over the 65 beads was used to compute the PSF and, after Fourier Transform, the OTF, that is used in the RL deconvolution.

## Simulation setup

We have used the "cameraman" stock image, which is $256 \times 256$ pixels. We assume the image is oversampled twice (pixel size $\lambda/(8NA)$). The image is normalized to average pixel value equal to one, then low pass

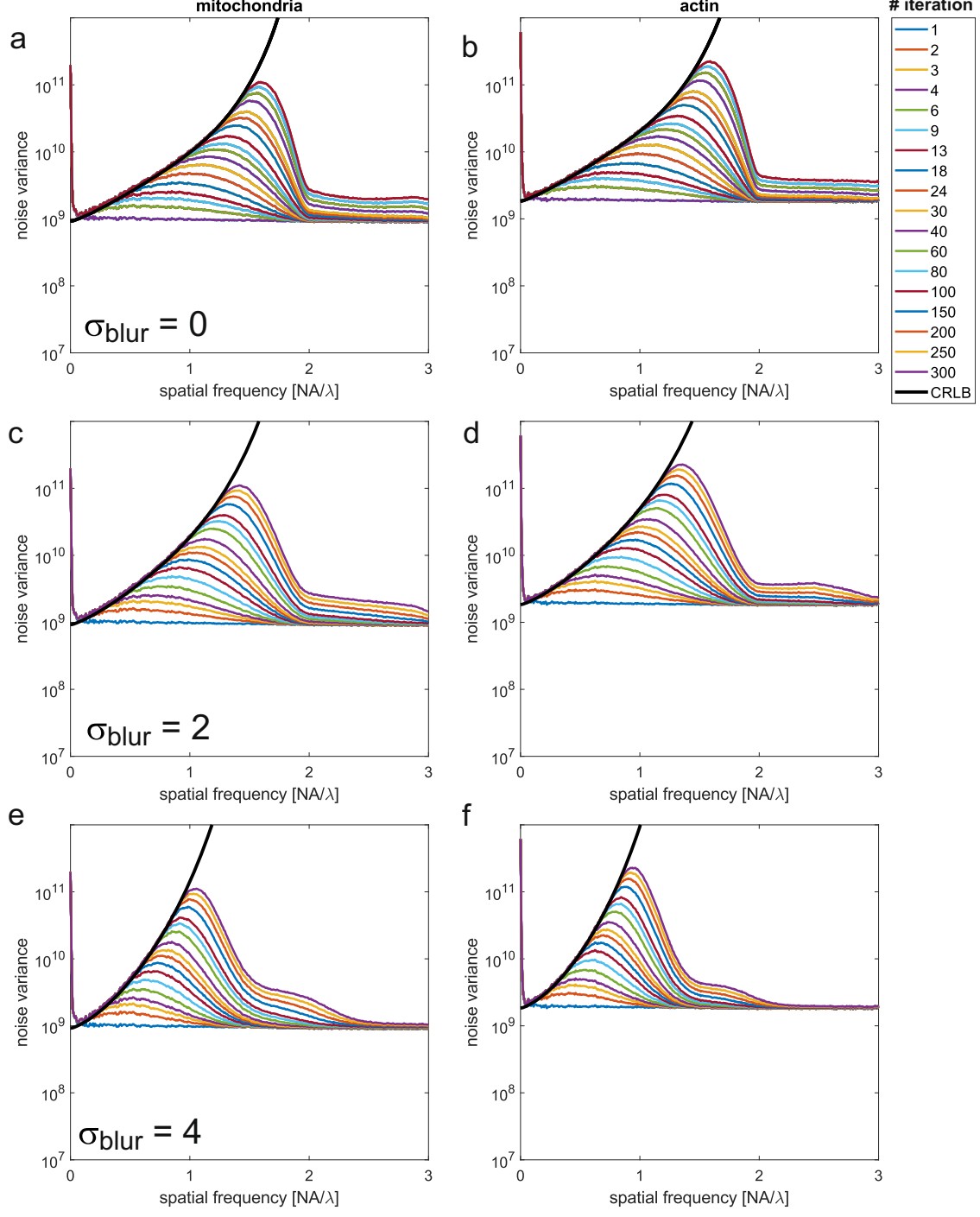

**Fig. 4 | Noise blow-up of RL-deconvolution in comparison to CRLB.**
**a** Noise variance averaged over rings in Fourier space for **a** the mitochondria channel, and **b** the actin channel, for the PSF/OTF model without additional blurring. **c**, **d** Same, for additional PSF/OTF blurring with a Gaussian with $\sigma_{blur} = 2$ pixels. **e**, **f** Same, for additional PSF/OTF blurring with a Gaussian with $\sigma_{blur} = 4$ pixels.

filtered with the scalar OTF as function of spatial frequency $q = |\bar{q}|$:

$$\hat{g}_{sc} = \frac{2}{\pi}\left[\arccos\left(\frac{q\lambda}{2NA}\right) - \left(\frac{q\lambda}{2NA}\right)\sqrt{1 - \left(\frac{q\lambda}{2NA}\right)^2}\right] \quad (4)$$

and multiplied with a factor $\mu_0 = 1000$ (average expected number of photons per pixel). Finally, noise is added according to Poisson statistics. In the deconvolution, an OTF $\hat{g}_{dec} = \hat{g}_{sc}{}^{\nu}$, where $\nu$ is an exponent quantifying the inaccuracy of the OTF, is assumed. The exponent is set to values $\nu = 1.0, 1.5, 2.0, 2.5$. The simulation is repeated $M = 10$ times for computing the noise variance in Fourier space.

### Reporting summary
Further information on research design is available in the Nature Portfolio Reporting Summary linked to this article.

### Data availability
Data is available at the 4TU.ResearchData repository https://doi.org/10.4121/b47dff09-f266-42fc-87af-3c7ad9349de7.

## Code availability

Matlab code is available at the GitLab repository https://gitlab.tudelft.nl/imphys/ci/rl-deconvolution-noise.

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

## Acknowledgements

Y.L. was supported by the Dutch Research Council (NWO), under project 19745, S.P. by the Dutch Research Council (NWO), under project 17046, Y.W. and S.S. acknowledge support by an ERC Advanced Grant, under grant agreement no. 101055013. We thank Andrew York for stimulating discussions on deconvolution.

## Author contributions

Simulations were done by Y.L, imaging experiments by S.P. and Y.W., data analyses were done by S.S. S.S. supervised the study and wrote the paper.

## Competing interests

The authors declare no competing interests.
