## [Transparent Peer Review file · Nature Communications]

Noise amplification and ill-convergence of Richardson-Lucy deconvolution

Corresponding Author: Professor Sjoerd Stallinga

Version 0:

Reviewer comments:

Reviewer #1

(Remarks to the Author)
Dear Authors,

The work in this manuscript addresses a well-known problem in fluorescence microscopy: deconvolution strategies form an essential part of many data processing pipelines, but when applying them, a careful balance needs to be struck since, eventually, unnatural and artifactual structures start appearing in the images. This paper gets at the heart of this problem by providing theoretical proof that the deconvolution of band-limited signals is not convergent by its very nature. To the best of my knowledge, it is the first time that such a meticulous approach has been applied to this question. I believe this publication should be published as soon as possible and will guide further developments in data analysis within microscopy while also providing foundations for related fields where data deconvolution is paramount (such as spectroscopy and image processing).

My main concern is that while the main findings are clearly outlined in the main text and illustrated amply using the added graphs and figures, The work in supporting information is much harder to read. Given the importance of the theoretical findings enclosed, I would strongly recommend adding an expanded, straightforward narrative text to accompany the derivation, making it understandable to a broader audience.

(Remarks on code availability)

Reviewer #2

(Remarks to the Author)

The authors derive Cramer Rao Lower Bound (CRLB), the minimum possible variance or mean squared error, for the images restored by the Richard-Lucy deconvolution (RL) algorithm that maximizes the Poisson product likelihood.

They demonstrate that this lower bound diverges (in the Fourier domain) for larger spatial frequencies within the diffraction-limited bandpass and with increasing image size, being inversely proportional to the square of the optical transfer function (OTF) and directly proportional to the image size.

The authors verify this divergence with experimental data.

This theoretical result is very important and confirms that the RL algorithm is problematic at its core and does not converge, resulting in commonly seen artifacts. Any appearance of convergence implies a local maximum.

The paper is mostly well-written and easy to read. I have minor comments that should be addressed before publication.

1. In equation 2, the authors show CRLB bound in real space and say that it diverges with image size K . However, in the equation, K appears in denominator on the right hand side of the inequality. Naively, this seems as if the bound would decrease with increasing K instead of diverging. Am I missing something?
2. In the main manuscript, it would be nice to mention what happens to the

bound for frequencies beyond the bandpass where OTF is zero. If the same formula as in equation 1 applies, it is clearly not well-defined owing to division by zero. On face value, this point seems trivial however people using RL deconvolution often misleadingly say that RL deconvolution can super-resolve based on deconvolution performed on the PSF itself, that is, on pointilistic or sparse samples. This would clarify that results beyond the bandpass cannot be trusted.

3. I am a little confused by the statement in the third paragraph on page 3 "noise blowup is not rooted in the ill-posedness of the deconvolution problem". I am guessing the authors here define ill-posedness strictly based on zero information available beyond the bandpass. However, at least, in my mind, decrease in OTF magnitude with spatial frequency implies reduction in contrast and consequently reduction in information. Therefore, the ill-conditioning of the solution is increasing with spatial frequency. In other words, information content progressively decreases instead of suddenly at the diffraction limit. A counterexample may be when the OTF is constant (hypothetically) and suddenly drops to zero at diffraction limit like a step function. Would the noise amplify in that case? Am I misunderstanding something?

4. We know that standard RL algorithm maximizes a Poisson product likelihood, so it is confusing to me why authors are discussing Poisson-Gaussian statistics in the results section on page 2. In the supplementary as well, a mixed Poisson-Gaussian distribution (equation 5) is used for the derivation. What is the purpose for this? Is it just to be more general and realistic? But in the supplementary equation 12, you eventually assume the Poissonian update rule by setting $\beta = x$.

5. In Figure 2 and 3, it would be easier for the reader to mark the diffraction limit with a circle in the Fourier transform plots.

6. Figure 4 is a little confusing. While I understand that noise variance is increasing with spatial frequency and iteration number demonstrating the amplification, why does CRLB seems like the upper bound in this figure instead of a lower bound. All the curves seem to lie below the CRLB curve instead of above. Am I misunderstanding something here? The significance of CRLB is not very clear from this figure.

I would definitely recommend publication upon clarification on these points.

Steve Presse

(Remarks on code availability)

Reviewer #3

(Remarks to the Author)

The contribution to the state-of-the-art methods is not accurately mentioned.

The authors should add a related work section, reviewing the different versions of the Richardson-Lucy (RL) algorithms and the different methods used for OTF estimation.

Concluding remarks on the reviewed methods are needed at the end of the review section.

In the main article, the authors did not mention which version of the RL algorithm has been utilized, and why this version is selected for this methodology.

the authors should provide a pseudo-code, explaining in steps the proposed methodology.

also, the authors should mention all the mathematical equations and define all the variables.

Some equations are stated without a reference. The authors should explain where or how they obtain such equations.

from the provided results and video files, I can see the results having ringing artifacts and a white shadow around the edges. this mainly indicates that the OTF estimation is inaccurate.

The OTF estimation should be refined in accordance with the iterations increase.

The experimental results are limited to proving the efficiency of the proposed model. Therefore, authors must present more results to demonstrate their method's effectiveness.

The proposed method should be compared with other developed versions of the RL algorithm, and the authors should justify why their method is better than the other methods. (the same should be applied to the OTF estimation method, i.e. two types of comparisons must be performed).

the authors should use advanced IQA metrics to measure the accuracy.

the authors should also measure the implementation times and the amount of used memory.

using the recoded times and memory usage, the authors should measure the complexity of the proposed method and the compared methods.

the future works are not presented clearly.

where do we use this work? kindly explain its real-life application.

Some linguistic errors are discovered.

(Remarks on code availability)

Version 1:

Reviewer comments:

Reviewer #1

(Remarks to the Author)

I want to express my gratitude to the authors for the additional clarifications provided in the supporting information and the notes added to the main text in response to reviewer #2's comments. These additions will greatly enhance clarity for non-expert users and increase the overall impact of the paper. I would also like to emphasize my belief that this work should be published as soon as possible, as it addresses a topic central to many of the current developments in imaging.

(Remarks on code availability)

Reviewer #2

(Remarks to the Author)

The authors have addressed all comments. Recommend publication.

(Remarks on code availability)

Reviewer #3

(Remarks to the Author)

(Remarks on code availability)

Dear editor

Enclose you find a revised version of our manuscript. We have improved it according to the different points of advice from the reviewers. Below you find detailed answers to the points raised by them. We want to thank the reviewers for their careful review. We believe that in its current form the manuscript is ready for publication in Nature Communications.

With kind regards, also on behalf of my co-authors,

Sjoerd Stallinga

#####

REVIEWER COMMENTS

Reviewer #1 (Remarks to the Author):

Dear Authors,

The work in this manuscript addresses a well-known problem in fluorescence microscopy: deconvolution strategies form an essential part of many data processing pipelines, but when applying them, a careful balance needs to be struck since, eventually, unnatural and artifactual structures start appearing in the images. This paper gets at the heart of this problem by providing theoretical proof that the deconvolution of band-limited signals is not convergent by its very nature. To the best of my knowledge, it is the first time that such a meticulous approach has been applied to this question. I believe this publication should be published as soon as possible and will guide further developments in data analysis within microscopy while also providing foundations for related fields where data deconvolution is paramount (such as spectroscopy and image processing).

Thank you.

My main concern is that while the main findings are clearly outlined in the main text and illustrated amply using the added graphs and figures, The work in supporting information is much harder to read. Given the importance of the theoretical findings enclosed, I would strongly recommend adding an expanded, straightforward narrative text to accompany the derivation, making it understandable to a broader audience.

Indeed, the derivation in the supporting information is rather dense. We have expanded the text around the math considerably, making it more accessible to the a wider readership.

Reviewer #2 (Remarks to the Author):

The authors derive Cramer Rao Lower Bound (CRLB), the minimum possible variance or mean squared error, for the images restored by the Richard-Lucy deconvolution~(RL) algorithm that maximizes the Poisson product likelihood. They demonstrate that this lower bound diverges (in the Fourier domain) for larger spatial frequencies within the diffraction-limited bandpass and with increasing image size, being inversely proportional to the square of the optical transfer functon(OTF) and directly proportional to the image size. The authors verify this divergence with experimental data. This theoretical result is very important and confirms that the RL algorithm is problematic at its

core and does not converge, resulting in commonly seen artifacts. Any appearance of convergence implies a local maximum. The paper is mostly well-written and easy to read.

Thank you.

I have minor comments that should be addressed before publication.

1. In equation 2, the authors show CRLB bound in real space and say that it diverges with image size K . However, in the equation, K appears in denominator on the right hand side of the inequality. Naively, this seems as if the bound would decrease with increasing K instead of diverging. Am I missing something?

The explanation around eq. 2 was rather sparse leading to this misunderstanding. Indeed, there is a factor K in the denominator, but the nominator is a sum over K terms. In other words, it is an average. As $K \rightarrow \infty$ this average becomes an integral, which formally diverges. We have now as accompanying text: "The term between brackets in the lower bound is the average of the inverse square of the OTF, and is much larger than one. As the image size grows large ($K \rightarrow \infty$) this average becomes an integral, which formally diverges (see Supplementary Information)."

2. In the main manuscript, it would be nice to mention what happens to the bound for frequencies beyond the bandpass where OTF is zero. If the same formula as in equation 1 applies, it is clearly not well-defined owing to division by zero. On face value, this point seems trivial however people using RL deconvolution often misleadingly say that RL deconvolution can super-resolve based on deconvolution performed on the PSF itself, that is, on pointillistic or sparse samples. This would clarify that results beyond the bandpass cannot be trusted.

This is very astute of the reviewer. We have wondered ourselves too about what could be said about the bound for spatial frequencies beyond the cutoff. Our conclusion is that we can actually say nothing about noise in this regime from a theoretical side. These out of band components of the object cannot be estimated as there is no information available. A CRLB for parameters that you cannot estimate is not a meaningful concept, hence there is no lower bound there. On the empirical side though, the actual noise profiles and spectral SNR results do point towards a clear conclusion. The out-of-band noise does increase with iteration number (see Figure 4) but quantitatively not so fast as the in-band noise. This suggests divergent noise for the iteration count going to infinity there too. Along the way though, for a limited number of iterations, there is an increase in spectral SNR also out of band (see e.g. Supplementary Figure 3 bottom row for high SNR raw data). So, retrieval out of band is possible, albeit limited, and not guaranteed for an arbitrary object structure. In ref. 8, Heintzmann makes this argument convincingly. Enhancement turns out to work for many "natural" images, where there are plenty of edges in the image. We added the following text: "It appears that SSNR can also be increased for spatial frequencies beyond the diffraction limit (see e.g. Supplementary Figure 3j). This SSNR increase drops to zero when the number of iterations goes to infinity, similar to the SSNR increase for the spatial frequencies below the diffraction limit. This points to a blowup of noise also for these higher spatial frequencies, as seems to be the case in experiment (see growth in noise level curves with iteration number in Figure 4), even though a formal CRLB cannot be established in the spatial frequency region beyond the cutoff."

3. I am a little confused by the statement in the third paragraph on page 3 "noise blowup is not rooted in the ill-posedness of the deconvolution problem". I am guessing the authors here define ill-posedness strictly based on zero information available beyond the bandpass. However, at least, in my

mind, decrease in OTF magnitude with spatial frequency implies reduction in contrast and consequently reduction in information. Therefore, the ill-conditioning of the solution is increasing with spatial frequency. In other words, information content progressively decreases instead of suddenly at the diffraction limit. A counterexample may be when the OTF is constant (hypothetically) and suddenly drops to zero at diffraction limit like a step function. Would the noise amplify in that case? Am I misunderstanding something?

The reviewer raises interesting points. Indeed, with “ill-posedness” we mean the zero information beyond the band limit. The formulation of the reviewer in which this is gradually approached is more clear and to the point, so we modified the formulation accordingly. The text now reads “... in the ill-posedness of the deconvolution problem for the spatial frequencies above the cutoff. Rather, the OTF gradually decreases with spatial frequency, and the noise amplification is strongest in the spatial frequency region approaching the cutoff. It is therefore also to be expected that noise amplification will occur in a hypothetical imaging system with a non-zero OTF for all spatial frequencies....”. The counterexample mentioned by the reviewer of a flat OTF until the cutoff is interesting from multiple perspectives. In this case noise blowup is not expected, as the CRLB gives a flat lower limit. Alas, the whole RL-deconvolution procedure would fail in this case too. There are two reasons for that. First of all, the flat OTF gives no push for the algorithm to sharpen up anything, so I expect that the algorithm would have zero to negligible impact. Second, a flat OTF until the cutoff violates the so-called Lukosz bound, an upper limit for the OTF so that the PSF remains positive. In the absence of the positivity constraint it becomes rather unpredictable what the RL-algorithm would do. We added a small discussion of this case: “Another hypothetical case of interest is that of a flat OTF until the cutoff. Then, noise amplification and blowup is not expected as the CRLB is also flat in spatial frequency space. The applicability of RL deconvolution to this case, however, is questionable, as the flatness of the OTF does not induce the algorithm to retrieve object spatial frequencies with low contrast, i.e. no sharpening effect can be expected. Furthermore, the flat OTF violates the Lukosz bound and gives rise to a non-negative PSF, which is problematic in view of the built-in positivity constraint of the RL deconvolution algorithm.”.

4. We know that standard RL algorithm maximizes a Poisson product likelihood, so it is confusing to me why authors are discussing Poisson-Gaussian statistics in the results section on page 2. In the supplementary as well, a mixed Poisson-Gaussian distribution (equation 5) is used for the derivation. What is the purpose for this? Is it just to be more general and realistic? But in the supplementary equation 12, you eventually assume the Poissonian update rule by setting $\beta = x$.

The standard RL algorithm (Eq. 12 supplementary information) also optimizes the mixed Poisson-Gaussian likelihood (Eq. 5 supplementary information). The purpose of including this effect is to take into account the realistic noise on images acquired with modern sCMOS cameras. These have small but non-zero readout noise. The impact of this, however, is limited because typically the shot noise variance (on average equal to mean # photons/pixel), is much larger than the readout noise variance. We mentioned this in the methods section (“We use the default RL-algorithm, with a slight modification to take the Gaussian camera readout noise into account as well (see Supplementary Information). The impact of this was rather small, as the typical shot noise variance (mean number of photons per pixel) was usually much larger than the readout noise variance (around 4 based on the above mentioned 1.93 e we measured).”), and expanded the treatment in the Supplementary information as well to bring across this point better.

5. In Figure 2 and 3, it would be easier for the reader to mark the diffraction limit with a circle in the Fourier transform plots.

This has been added to the Figures.

6. Figure 4 is a little confusing. While I understand that noise variance is increasing with spatial frequency and iteration number demonstrating the amplification, why does CRLB seems like the upper bound in this figure instead of a lower bound. All the curves seem to lie below the CRLB curve instead of above. Am I misunderstanding something here? The significance of CRLB is not very clear from this figure.

This is indeed a subtle point. The discussion of Figure 4 has been expanded to explain this better: "There, we see the buildup of noise with iteration number, starting from the flat noise profile of the raw image data before the first iteration, which is well below the CRLB. The noise level must rise to at least the level of the CRLB when the likelihood increases to its maximum. The experimental curves indicate that this situation is gradually approached as the number of iterations grows, with a noise level at the CRLB for a gradually growing region in spatial frequency space." We hope this makes matters more clear.

I would definitely recommend publication upon clarification on these points.
Steve Presse

Reviewer #3 (Remarks to the Author):

We want to thank also this reviewer for their comment. In many remarks, however, the intent of the reviewer is not so clear and we are at a loss how exactly to improve the manuscript according to the reviewer's advice. We nevertheless made a best effort to do so. In several remarks we think the reviewer may have misunderstood the goal of this study. It is by no means a comparative study between different forms of and variations on the Richardson-Lucy (RL) algorithm, let alone that we provide an exhaustive overview of these. Instead, the goal is to reveal the root cause of an essential shortcoming of the RL algorithm, namely its sensitivity to noise and its ill-convergence.

The contribution to the state-of-the-art methods is not accurately mentioned.
The authors should add a related work section, reviewing the different versions of the Richardson-Lucy (RL) algorithms and the different methods used for OTF estimation.

We kindly refuse to do so. As stated above, the goal of the paper is the study of noise amplification and ill convergence, not comparing different implementations of RL or different ways to estimate the OTF. In the introduction we give a fair overview of relevant prior works, but this of course not an exhaustive overview of the huge body of work in the field of deconvolution. We also think that would not be fitting with the stated goals of the paper, and is hence unreasonable to add. In our image data processing we use an established method for OTF estimation, namely fitting a through-focus bead stack (see e.g. ref 30). We do not aim for a comparison of experimental OTF estimation methods, but simply use one which is known to work fairly well.

Concluding remarks on the reviewed methods are needed at the end of the review section.

We do not understand which section the reviewer means here.

In the main article, the authors did not mention which version of the RL algorithm has been utilized, and why this version is selected for this methodology. The authors should provide a pseudo-code, explaining in steps the proposed methodology.

We use the standard version of the RL algorithm, as explained in the supporting information. We added a remark that we do so in the methods section of the manuscript: "We use the default RL-algorithm, with a slight modification to take the Gaussian camera readout noise into account as well (see Supplementary Information). The impact of this is rather small, as the typical shot noise variance (mean number of photons per pixel) is usually much larger than the readout noise variance (around 4 based on the above mentioned 1.93 e we measured).", see also point with reviewer 2. Pseudo-code seems superfluous for such an established method so this is not added.

Also, the authors should mention all the mathematical equations and define all the variables. Some equations are stated without a reference. The authors should explain where or how they obtain such equations.

We believe we have done so already. Without specific pointers from the reviewer we do not know what to correct.

From the provided results and video files, I can see the results having ringing artifacts and a white shadow around the edges. This mainly indicates that the OTF estimation is inaccurate. The OTF estimation should be refined in accordance with the iterations increase.

As stated earlier, the goal of the paper is not to provide a best possible RL estimate with an OTF estimate that is as good as can be. In Figure 4 we show noise blowup in cases where we deliberately modify the experimentally retrieved OTF for application in the RL algorithm, i.e. the OTF is inaccurate by design. In all cases there is noise blowup following the diverging CRLB. This is supported by the simulation results in Movie 4. So, any inaccuracies in the OTF do not alter the conclusion of our work, namely, that noise will be amplified as the iteration continues. Furthermore, we believe that the actual OTF used on the experimental data is rather accurate (see Supplementary Figure 1 and Movie 3 which compares the measured through-focus PSF with the fitted through-focus PSF).

The experimental results are limited to proving the efficiency of the proposed model. Therefore, authors must present more results to demonstrate their method's effectiveness.

We do not know what the reviewer means with the efficiency of the proposed model. Again, we do not propose a new model or new way to do RL deconvolution.

The proposed method should be compared with other developed versions of the RL algorithm, and the authors should justify why their method is better than the other methods. (the same should be applied to the OTF estimation method, i.e. two types of comparisons must be performed). The authors should use advanced IQA metrics to measure the accuracy.

As stated before, we are not proposing a new model, nor do we claim to do anything better than available in the state of the art. So, we cannot follow up on the reviewer's recommendation.

The authors should also measure the implementation times and the amount of used memory. Using the recorded times and memory usage, the authors should measure the complexity of the proposed method and the compared methods.

Computation times and memory load are the same as for standard RL deconvolution, and not the topic of this paper.

The future works are not presented clearly.

We do not know what exactly is unclear to the reviewer.

Where do we use this work? kindly explain its real-life application.

Again, we do not propose a new method, but elucidate the shortcomings of an existing method. So, the goal was to answer a scientific question, which we believe we convincingly did. The outlook statement in the discussion section points to different other computational imaging modalities (phase imaging, ptychography, structured illumination microscopy) where a similar CRLB analysis could be tried.

Some linguistic errors are discovered.

It is not clear which errors the reviewer means.